# Psychometric Characteristics of the Self-Care of Chronic Illness Inventory in Older Adults Living in a Middle-Income Country

**DOI:** 10.3390/ijerph20064714

**Published:** 2023-03-07

**Authors:** Alta Arapi, Ercole Vellone, Dhurata Ivziku, Blerina Duka, Dasilva Taci, Ippolito Notarnicola, Alessandro Stievano, Emanuela Prendi, Gennaro Rocco, Maddalena De Maria

**Affiliations:** 1Department of Biomedicine and Prevention, University of Rome Tor Vergata, 00133 Rome, Italy; 2Department of Nursing and Obstetrics, Wroclaw Medical University, 50-367 Wroclaw, Poland; 3Degree Course in Nursing, UniCamillus International Medical University, 00131 Rome, Italy; 4Department of Healthcare Professions, Campus Bio-Medico University Hospital Foundation, 00128 Rome, Italy; 5Degree Course in Nursing, Catholic University “Our Lady of Good Counsel”, 1000 Tirana, Albania; 6Centre of Excellence for Nursing Scholarship, Order of Nurses of Rome, 00173 Rome, Italy; 7Department of Clinical and Experimental Medicine, University of Messina, 98100 Messina, Italy; 8Department of Biomedical Sciences, Catholic University “Our Lady of Good Counsel”, 1000 Tirana, Albania

**Keywords:** self-care, chronic illness, older adults, instrument, psychometric testing, validity, reliability

## Abstract

Chronic illness requires numerous treatments and self-care is essential in the care process. Evaluation of self-care behaviors facilitates the identification of patients’ needs and optimizes education and care processes. This study aimed to test the psychometric characteristics (validity, reliability, and measurement error) of the Albanian version of the Self-Care of Chronic Illness Inventory (SC-CII). Patients with multiple chronic conditions and caregivers were recruited in outpatient clinics in Albania. The patients completed the SC-CII, which includes three scales: self-care maintenance, self-care monitoring, and self-care management. Factorial validity was tested for each scale, with confirmatory factor analysis. Reliability was evaluated with the composite coefficient, Cronbach’s alpha, and the global reliability index for multidimensional scales. The construct validity was tested using hypothesis testing and known differences between groups. The measurement error was tested to assess responsiveness to changes. The self-care maintenance and self-care monitoring scales showed a unidimensional factorial structure, while the self-care management scale showed a bidimensional structure. Reliability estimates were adequate for all reliability coefficients. Construct validity was supported. The measurement error was adequate. The Albanian version of the SC-CII shows good psychometric properties in the Albanian sample.

## 1. Introduction

Older people often suffer simultaneously from more than two chronic illnesses, defined as multiple chronic conditions (MCCs) [1]. These have increased in recent years, becoming responsible for 74% of all deaths worldwide, with an important increase in the rate in low and middle-income countries as well [2]. MCCs represent an important social burden [3] due to the high morbidity and disability, and negatively affect patients’ and their family members’ quality of life [4]. Furthermore, MCCs are a global concern for healthcare systems because they are associated with substantial increases in healthcare costs and resource utilization [5]. According to the WHO, MCCs account for 11% of premature deaths that occur in low/middle-income countries (LMIC), including Albania [6]. As such, the WHO recommends the practice of self-care behaviors in the management of MCCs to mitigate worse outcomes associated with them, not only in industrialized countries but also in LMIC [6].

Self-care is a process of maintaining health status through the promotion of health and disease management practices [7]. Associations between adequate self-care practices and reduced hospitalizations or mortality rates and better quality of life or well-being among patients are well documented [8]. Nevertheless, self-care behaviors are frequently inadequate due to the age of the patient, comorbidities, cognitive impairments, stressful life events, and the influence of others [8,9]. Furthermore, culture, social norms, values, meanings, language, environments, attitudes, behaviors, personal perceptions, and care partner [10] can influence self-care behaviors [7,11]. Thus, healthcare professionals should support and empower patients with chronic illnesses and their care partners to perform self-care. This suggests the need for a regular assessment of self-care behaviors to identify inadequate caring standards and implement customized interventions.

Albania is a small middle-income country in the southeast of Europe. In recent years, the country has been exposed to deep political and socioeconomic reforms, which caused important epidemiological and health changes and a significant increase in chronic diseases and MCCs [12]. In fact, chronic diseases account for 89% of total deaths in the country [13], with a 45% increase in the prevalence of MCCs [14], including cardiovascular diseases, diabetes, and chronic respiratory diseases [15]. The “modernization” process [12] nurtured risky health behaviors such as increased tobacco and alcohol consumption, sedentary lifestyles, and poor dietary habits. All these factors require Albanians to adopt self-care behaviors that need to be properly assessed to implement interventions aimed at improving them.

Furthermore, Albanians inherited the culture of ‘curing’ rather than ‘preventing’, the overuse of hospital services, and the lack of awareness of MCCs from the previous regime [13]. In addition, the nation is facing demographic challenges such as young adults’ migration to other nations or to urban areas, and smaller family nuclei [16]. This sociodemographic aspect might influence the self-care abilities of elderly people with MCCs in Albania. Collectively, these increasing trends in unhealthy behaviors, reduced health literacy, the poor culture of prevention, and demographic changes suggest the urgent need for intervention in these amendable risk factors [12] to reduce the increasing burden of MCCs in the Albanian population [16].

In the last decade, the government has employed different approaches to improve primary healthcare services such as the implementation of free check-ups, free medical visits in primary healthcare departments, and reimbursement of medications for chronic illnesses [16]. Consequently, some progress has been documented on the culture of prevention around chronic disease [13]. However, despite these efforts, self-care behaviors in the Albanian population affected by MCCs are underexplored. The few data available document that 59% of hospitalized patients with chronic diseases (diabetes mellitus, heart diseases, hypertension, chronic lung diseases, osteoarthritis, bronchial asthma, and chronic kidney disease) are not able to recognize the risk factors, monitor their status, and control the progression of the disease [15]. Therefore, in Albania, long-term effort is requested from healthcare professionals to assess abilities and to educate patients with MCCs and their care partners on self-care practices to manage the chronic diseases [15,16].

An instrument internationally used to assess self-care in chronic diseases is the Self-Care of Chronic Illness Inventory (SC-CII). The SC-CII [17] was developed in English and translated into many languages such as Arabic, Catalan, Chinese, Dutch, Italian, Spanish, and Swedish, but not Albanian [18]. It captures the behaviors of the self-care process with three separate scales: self-care maintenance, self-care monitoring, and self-care management. Self-care maintenance refers to behaviors performed to improve well-being, maintain health, or maintain physical and emotional stability (e.g., “take medications as prescribed”) [7]. Self-care monitoring is the surveillance of chronic conditions (that is, “monitoring signs and symptoms”); this process involves the evaluation and perception of bodily changes by listening to the body [7]. Self-care management includes the recognition of chronic disease signs and symptoms and the patient behaviors in response to such symptoms (e.g., “take medicines to make the symptom decrease or go away”) [7]. The SC-CII has shown adequate validity and reliability across populations with supportive fit indices in confirmatory factor analysis (e.g., comparative fit index (CFI) ranged between 0.93 and 1.00 in the three scales) and reliability (reliability coefficients for all the three scales ranged from 0.67 to 0.86) [7,18]. To date, no valid and reliable instrument is available to assess self-care in MCCs in an LMIC, such as Albania, and consequently, self-care behaviors cannot be assessed. A valid and reliable psychometrically sound instrument would allow not only an accurate assessment of self-care, but also evaluations of interventions aimed at improving self-care in these populations.

This study aimed to test the psychometric characteristics (validity, reliability, and responsiveness to changes) of the instrument Self-Care of Chronic Illness Inventory, in older Albanian adults affected by chronic illnesses.

## 2. Materials and Methods

### 2.1. Design

For this study, we used a cross-sectional multicenter design conducted in Albania.

### 2.2. Sample and Setting

A sample of 250 patients, considered adequate for these analyses [19], was recruited in outpatient clinics and community healthcare settings in central and south Albania. Inclusion criteria were age ≥ 65 years old, and a simultaneous diagnosis of chronic diseases such as heart failure, diabetes mellitus, chronic obstructive pulmonary disease, and at least one other chronic disease. Patients presenting a diagnosis of cancer or dementia were excluded.

### 2.3. Data Collection

Data were collected in the central and south Albanian regions between August 2020 and April 2021 with face-to-face interviews conducted by trained nurse research assistants. A paper survey was purposefully developed by the researchers for the study. To assure the quality of data, random data monitoring was performed by the principal investigator (MDM).

### 2.4. Measurements

The Self-Care of Chronic Illness Inventory (SC-CII) [20] is a self-report instrument based on the middle range theory of self-care of chronic illness [7]. The self-care maintenance scale has 7 items, the self-care monitoring scale has 5 items, and the self-care management scale has 6 items. Since the self-care management scale assesses responses to symptoms, this scale can only be completed if patients have symptoms. As recommended by Riegel and colleagues [21], for item #14, we used a 5-point ordinal response scale (from 1 “not quickly” to 5 “very quickly”) to form an observed variable. Each SC-CII item is measured using a 5-point Likert scale ranging from “Never” (1) to “Always”(5) (20). The three scales use a standardized score from 0 to 100, with higher scores indicating better self-care. The cut-off point for self-care adequacy is 70. The translation of the original English version to the Albanian version followed the Principles of Good Practice for the Translation and Cultural Adaptation Process for Patient-Reported Outcomes (PRO) measures [22].

The Self-Care Self-Efficacy Scale (SCSES) [23] was used to measure self-efficacy for self-care in chronic illness. It consists of 10 items with a total score ranging from 0 to 100, with higher scores indicating better self-care self-efficacy. The SCSES has been validated in different cultural groups, showing a supportive validity.

The 12-item Short Form Health Survey (SF-12) [24] version 2 was used to measure health-related quality of life (HRQOL). The scale is composed of two components of HRQOL: the physical (PCS) and mental (MCS). Each component scoring ranges from 0 to 100, where the higher the scores, the better the HRQOL. SF-12 reliability was tested with Cronbach’s alpha, which resulted in a coefficient of 0.84 for the PCS and 0.70 for the MCS [25]. SF-12 has been extensively used in patients suffering from chronic conditions [26,27].

The 9-item Patient Health Questionnaire (PHQ-9) [28] was used to measure depressive symptoms. This scale is composed of nine items with a 0–3 scoring possibility. The total score goes from 0 to 27; a higher score indicates higher depressive symptoms. The PHQ-9 showed good psychometric scoring for cutoff point ≥ 10: Cronbach’s alpha (0.86–0.89), test–retest (0.84), sensitivity and specificity (0.88) The PHQ-9 is available in many languages, and, in Albania, the tool is also used in primary healthcare centers to diagnose people with mental health conditions.

The one-item Dyadic Symptom Management Type Scale (DSMT) [29] was included to explore the organization and sharing of care activities within the patient–caregiver dyad. This scale identifies 4 types of dyadic management for chronic disease. When the patients’ and caregivers’ answers are concordant, the typologies are: (1) patient-oriented dyadic care type, where the patient performs the greatest part of self-care; (2) caregiver-oriented dyadic care type, where the caregiver performs the greatest part of self-care; and (3) collaborative-oriented type, where the patient and caregiver collaborate or complement each other in an equal manner. When the patient and caregiver provide a discordant answer, the dyad is classified as incongruent [30].

A sociodemographic questionnaire was used to collect sociodemographic characteristics and clinical data of participants, such as age, gender, education level, marital status, family income, and employment status, and the number and the type of the chronic diseases.

### 2.5. Ethical Considerations

Ethical approval for the study was obtained from the Catholic University of Our Lady of Good Counsel with protocol number 237/2020. The study was carried out according to ethical standards and according to the principles of the Declaration of Helsinki [31]. All participants received adequate information regarding the study and afterwards were asked to sign the informed consent form.

### 2.6. Statistical Analysis

Descriptive statistics (mean, standard deviation, percentages, and frequencies) were calculated out to describe the sample characteristics and SC-CII items. Skewness and kurtosis univariate indices were considered to evaluate the normal distribution of the items.

As dimensionality testing preceded reliability testing [32], we began the psychometric analysis of SC-CII performing confirmatory factorial analysis (CFA), and then tested its reliability. Consistent with recent recommendations about self-care inventory validation studies [20,33,34,35], we performed three separate CFAs, one for each scale (self-care maintenance, self-care monitoring, and self-care management). A general model with all three SC-CII scales was tested as well, similar to previous self-care inventory validation studies [20,34,36]. For the CFAs, we tested the same factorial structure tested by Riegel et colleagues (2019) [34]. Specifically, for the self-care maintenance scale, we specified two factors as follows: ‘health-promoting behaviors’ (items #1, #3, and #8) and ‘illness-related behaviors’ (items #2, #4, #5, and #6). Consistent with previous validation studies of SC-CII [20,34,37], item 7 (‘avoiding tobacco smoke’) was excluded from our analyses. Regarding the self-care monitoring scale, we specified a unidimensional factor model including items 9 through 13. Finally, with respect to the self-care management scale, we specified two factors: ‘autonomous behavior’ (items #14, #15, #16, and #20) and ‘consulting behaviors’ (items #17, #18, and #19) [20,34,36]. Due to the non-normal distribution of SC-CII items, the maximum likelihood robust (MLR) estimator [38] was used for parameter estimation.

Different fit indices were tested in the CFA: the comparative fit index (CFI), Turker and Lewis index (TLI), root mean square error of approximation (RMSEA) and standardized root mean square residual (SRMR) [39,40]. The goodness of fit values were interpreted following the literature recommendations [41,42].

Consistent with previous validations of self-care inventories [33,35], reliability was calculated with the composite reliability, or omega coefficient, that is indicated for multidimensional scales [43]. For completeness, the Cronbach alpha coefficient was calculated. Considering that the SC-CII is composed of several dimensions, we computed the global reliability index as well, which is specific for multidimensional scales [44]

Construct validity was tested via hypothesis testing, following Terwee’s recommendations [19] and known group differences. Specifically, we hypothesized that the SC-CII scale scores were significantly correlated with the Self-Care Self-Efficacy Scale, SF-12 (MCS and PCS), and PHQ-9 scores. Construct validity was further verified by posing the hypothesis that male patients with MCCs would have a higher score in self-care monitoring and self-care management behaviors than female patients [9], and the scores in all three scales of the SC-CII would be significantly higher in the patient-oriented dyadic care type among the typologies identified, with the patient being the major provider of self-care [45]. To test the associations with SC-CII scores, we used Pearson’s product–moment correlation coefficients with a significant *p* value set at <0.05. Correlations of 0.10–0.29 were considered small, 0.30–0.49 as moderate, and >0.50 as strong [46]. Differences between scores were identified through the T-test conducted using two different groups (sex and dyadic care type).

Additionally, we tested the SC-CII scale measurement error with the standard error of measurement (SEM) and the smallest detectable change (SDC). These values add additional information regarding the precision of an instrument and responsiveness to changes. To measure SEM, we used the formula standard deviation (SD) √ (1—reliability coefficient) [47], where the SD was the SD of the SC-CII scale score, and the reliability coefficient was the factor score determinacy coefficient or the global reliability index for multidimensional scales. A value of SEM < SD/2 indicates a more precise instrument. To measure the SDC, we used the formula 1.96 X√2 X SEM [48]. Smaller values of SEM and SDC indicate more precision in the instrument. Analyses were performed with SPSS 26 (IBM Corp., Armonk, NY, USA) and Mplus 8.2 (Los Angeles, CA, USA) [38].

## 3. Results

### 3.1. Sample Description

Table 1 presents the sociodemographic and clinical characteristics of the sample. On average, the patients were 73 years old, mostly male (53.6%), married (78.4%), retired (94.4%), and with a low level of education (62%). Most (80%) of them perceived that they earned enough to live, and they were living with a spouse (78%) and/or children (50%). The mean number of chronic illnesses was 2.5 (±0.69). The patients were mainly affected by hypertension (87.6%), diabetes mellitus (74.5%), and heart failure (44.8%).

### 3.2. Item Descriptive Analysis and Scale Scores

Table 2 reports the descriptive analysis of the SC-CII items. The sample reported a mean score of 72.0 (±16.0) for self-care maintenance, 75.0 (±19.1) for self-care monitoring, and 72.4 (±15.9) for self-care management. Patients with adequate self-care maintenance, self-care monitoring, and self-care management behavior made up 53.5%, 58.4%, and 53% of the sample, respectively.

Item 14 (“If you had symptoms in the past month, how quickly did you recognize it as a symptom of your illness”) was excluded by the dimensionality testing, as 4.7% of the sample did not present symptoms in the last 30 days. Among patients reporting chronic disease symptomatology, 3.2% specified that they were not able to recognize the symptom. Those who recognized the symptom referred different rapidity of recognition: 34.6% did not recognize it very quickly (score 1 or 2), 27.5% recognized it fairly quickly (score 3), and 29.9% recognized it quickly or very quickly (score 4 or 5).

### 3.3. Psychometric Analysis of the Self-Care Maintenance Scale

#### 3.3.1. Dimensionality

Since self-care maintenance is described as comprising “health promoting behaviors” and “illness-related behaviors” [17], we specified a two-factor confirmatory model. The model presented a misfit caused by the correlation between the two factors > 1. For this reason, we respecified a one-factor model and the goodness-of-fit indices of this model were inadequate. The model misfit was caused by an excessive shared variance between items #8 “Manage stress” and #4 “Eat a special diet” and between item #8 and #3 “Do physical activity”.

Since in people with chronic diseases, adherence to a diet can often be a source of stress in maintaining well-being, while the practice of physical activity is often used to manage stress, we specified the covariance between dietary adherence (item #4) and physical activity (item #3). Thus, we reran the model that yielded the following supportive fit indices: χ^2^ (12, N = 251) = 22.688, *p* = 0.0305, CFI = 0.967, TLI = 0.942, RMSEA = 0.060 (90% CI 0.018 0.097), *p* = 0.300, SRMR = 0.038. All factor loadings were significant (Figure 1, panel a).

#### 3.3.2. Scale Internal Consistency Reliability

The internal consistency reliability of the self-care maintenance scale was adequate. Specifically, the composite coefficient of the scale was 0.87 and the Cronbach’s alpha coefficient was 0.76.

### 3.4. Psychometric Analysis of the Self-Care Monitoring Scale

#### 3.4.1. Dimensionality

A one-factor model was tested that produced a partially adequate fit. The cause of the misfit was due to the excessive covariance between items #12 “Monitor whether you tire more than usual doing normal activities” and #13 “Monitor for symptoms”. The proximity of these items increased the shared variance between these two items [49]. For this methodological reason, we tested a model correlation between the residuals of these two items [50,51]. The model yielded an excellent fit as follows: χ^2^ (4, N = 251) = 4.268, *p* = 0.371, CFI = 0.999, TLI = 0.998, RMSEA = 0.016 (90% CI = 0.000 0.098), *p* = 0.648, SRMR = 0.014. All factor loadings were significant (Figure 1, panel b).

#### 3.4.2. Scale Internal Consistency Reliability

The internal consistency reliability, tested using the composite coefficient, for the self-care monitoring scale was high, 0.83, attesting to the internal coherence of the items. When the Cronbach alpha coefficient was computed, a coefficient of 0.88 was obtained.

### 3.5. Psychometric Analysis of the Self-Care Management Scale

#### 3.5.1. Dimensionality

Since self-care management is described as comprising autonomous behaviors and consulting behaviors [17], we tested a two-factor confirmatory model, The model yielded a poor fit. The misfit was caused by excessive covariance between items #17 “Take a medicine to make the symptom decrease or go away” and #18 “Tell your healthcare provider about the symptoms at the next office visit”. Additionally, in this case, as for the self-care maintenance scale, the proximity effects of these items could have produced an increase in the shared variance. We allowed the correlation between item residuals [50,51], and the fit indexes of the new model improved: χ^2^ (7, N = 226) = 10.027, *p* = 0.20, CFI = 0.985, TLI = 0.968, RMSEA = 0.044 (90% CI 0.000 0.099), *p* = 0.506, SRMR = 0.035. All factor loadings were significant, and two factors correlated 0.483 (*p* < 0.001), (Figure 1, panel c).

#### 3.5.2. Scale Internal Consistency Reliability

The internal consistency reliabilities of the two self-care management factors, tested using composite coefficients, were 0.79 and 0.67 for autonomous behaviors and consultive behaviors, respectively. The Cronbach’s alpha coefficient calculated for the entire six-item scale was 0.70. The global reliability index was 0.74 for the overall self-care management scale.

### 3.6. Construct Validity through Hypothesis Testing

The SC-CII scale scores correlated significantly with other measures supporting the construct validity of the instrument (Table 3). The self-care maintenance, monitoring, and management scales were significantly (*p* < 0.001) correlated with the self-care self-efficacy scores, depression, and mental and physical quality of life. Male patients with MCCs scored higher in self-care maintenance (*p* = 0.004) and self-care monitoring (*p* = 0.002) behaviors than female patients, but not in management behaviors. In the patient-oriented dyadic care type, patients scored higher in the self-care maintenance (*p* = 0.002), monitoring (*p* = 0.003), and management (*p* = 0.004) scales than the caregiver-oriented dyadic care type. (Table 4)

### 3.7. Measurement Errors of the SC-CII

The SEM of the SC-CII was 5.77, 7.88, and 8.11 for the self-care maintenance, self-care monitoring, and self-care management scales, respectively. These measures were considered adequate since the SEM values were <SD/2. The SDC was 6.66, 7.78, and 7.89 for the self-care maintenance, self-care monitoring and self-care management scales, respectively.

## 4. Discussion

This study aimed to test the psychometric properties (dimensionality, construct validity, internal consistency reliability, and measurement error) of the SC-CII in a middle-income population. The results show adequate validity and reliability, as shown in other previous studies of psychometric validation [18,20]. To our knowledge, this is the first study to test SC-CII in a southeastern European country, with important scientific and practical implications.

### 4.1. Dimensionality

Regarding the self-care maintenance scale, the CFA models reported in the literature were not confirmed in our population. In fact, while in the existing model, behaviors fitted within two dimensions, health promotion and illness-related behaviors [17], in the Albanian model, we found a one-factor solution. Additionally, the initial one-factor solution did not fit the data well, but when we allowed the covariance between the residuals of items #8 “Manage stress” and #4 “Eat a special diet” and items #8 and #3 “Do physical activity” to freely correlate, the model improved. Our interpretation of the differences that emerged is that, regarding self-care maintenance behaviors, older Albanian adults with MCCs, in contrast with older adults with MCCs in Western countries, might not distinguish between illness-related behaviors and health promotion behaviors. This can be explained by a plausible influencing effect of cultural and social characteristics of this population that are different from those of Western nations due to previous political choices and the history of the country. Leininger’s culture care theory describes existing care diversities and universalities across cultures, and recognizes the influences of historical, cultural, and social structure factors on health/wellness patterns and well-being, care meanings, expressions, or patterns [52]. More specifically, self-care in general and self-care maintenance behaviors were recently found to be influenced by cultural beliefs and social norms [53]. This study is testing the SC-CII for the first time in a middle-income country, and provides evidence that this different dimensionality can be culturally based. Further studies in other similar countries might support or hinder this finding. Regarding the common variance of residuals among items #3, #4, and #8, the specification of correlation of residuals is methodologically acceptable if it does not influence the other model parameters [54], as was the case in our model. The correlation between these items might indicate that in Albania, patients with MCCs associate “eating a special diet” and “physical activity” with stress in maintaining a healthy lifestyle, which can be explained with the reduced health literacy and an inadequate culture of prevention in this population [13,16].

The self-care monitoring scale measures patients’ observation of signs and symptoms of chronic conditions. In the self-care process, the recognition of symptoms of chronic diseases is essential for the management of symptoms and the ability to properly manage the disease [21]. One factorial model was tested, and psychometric findings were consistent with previous studies [18,20]. This suggests that the self-care monitoring behaviors explored by this scale seem to be interpreted and applied similarly by adults with MCCs across cultures and nationalities. In the model, we allowed for a covariance of residuals of items #12 “Monitor whether you tire more than usual doing normal activities” and #13 “Monitor for symptoms”. This covariance can be explained by two plausible motivations. First, the closeness of items in the scale might have influenced their covariance [49,51]. Additionally, we can hypothesize that, in the Albanian patients, tiredness seems to be a symptom that is easy to identify, and it is likely considered as a common symptom of the chronic diseases. In previous linguistic validation models of the scale, different item covariance residuals were allowed to correlate to improve the model fit. For example, in the Italian population, similar to that of the USA, residuals of items #9 and #10 were allowed to covariate, while in the Swedish population, this was applied to items #9 and #11 [18]. This suggests that symptom identification, monitoring, and association with the chronic illness can be subject to cultural influences. In fact, several studies have emphasized that culture influences symptom identification and monitoring [53]. Despite these item covariations, we can confirm that patients conceptualize the self-care monitoring in a similar way; they take actions related to the monitoring of the disease, but attribute different importance to scale items.

The self-care management scale comprises patients’ behaviors in response to symptoms of a chronic disease and incorporate behaviors in two dimensions: “autonomous behavior” and “consulting behavior” [20]. Autonomous behaviors refer to patients’ spontaneous actions to relieve symptoms based on prior experience, whereas consulting behaviors include adapting recommendations from others. We tested a two-factor confirmatory model, and the psychometric findings were consistent with previous studies [18,20]. Similar to self-care monitoring, self-care management behaviors in adults with MCCs are universally interpreted. How people deal with signs and symptoms of a disease can be related to their level of education, to their confidence in the recommendations of the healthcare professionals, to cultures, or simply to concerns about their health status [13], and behaviors are similar independent from culture or nationality. We allowed the covariance of residuals between items #17 “Take medicines to make the symptom decrease or go away” and #18 “Tell your healthcare provider about the symptom at the next office visit”. The shared variance is probably explained by the proximity effects of these items [49,51]. Additionally, in the Albanian sample, item #17 loaded in the “autonomous behavior” factor, confirming similar findings from a previous study [18]. This suggests that, in the Albanian population, some patients tend to use autonomous behaviors towards self-medication, while other patients refer to primary healthcare physicians to initiate or continue treatments. Another study documented similar behaviors among Albanian patients [16]. This is in line with the culture of ‘curing’ rather than ‘preventing’ inherited from the previous regime [13]. Despite their low level of education, Albanian patients showed themselves to be able to identify symptoms, autonomously take medications to relieve symptoms, and to refer behaviors to healthcare professionals during routine visits.

### 4.2. Scale Internal Consistency Reliability

The SC-CII presented good internal consistency reliability, with coefficients being higher than 0.70. This suggests that, despite the multidimensionality of the scale, the items reflect the same constructs, and can be combined into an overall score. Thus, our results indicate that the Albanian version of the SC-CII presents adequate reliability both at the factor and scale levels, indicating that the three SC scales are precise in measuring self-care (maintenance, monitoring, and management) behaviors in the multiple chronic care conditions in the sample studied.

### 4.3. Construct Validity Testing

The Albanian SC-CII presented a good construct validity, as supported by the presence of positive associations between these scales and the Self-Care Self-Efficacy Scale, as postulated by theories [7,55] and established previously in single chronic illness studies [33,35,56,57] and in MCCs [58]. Furthermore, we found that the higher self-care scores were associated with a better physical and mental quality of life and lower levels of depression scores than found in previous studies [59,60]. Regarding gender influences on the self-care behaviors, we partially confirm our hypothesis that male patients performed more self-care monitoring and management than females; we found statistically significant differences in self-care maintenance and monitoring, but not in self-care management scores. We suggest that male patients are more attentive regarding healthy behaviors and observation of sign and symptoms than female ones, but regarding the management of symptoms, they present similar behaviors. Our data are consistent with a previous study that showed that men and women alike seek explanations for bodily changes and take appropriate and timely action to manage signs and symptoms [61]. Finally, consistent with the classification system of patient–caregiver dyads proposed by Buck et al. (2019) [30], we found that in the patient-oriented dyadic typology, scores were higher in all the SC-CII scales when compared to the other dyadic typologies. This confirms that SC-CII is sensitive enough to capture differences in the levels of self-care maintenance, monitoring, and management.

### 4.4. Measurement Errors of the SC-CII

Measurement error testing is an important test to perform when validating scales. Regarding the SC-CII, SEM and SDC testing reported small scores. In fact, SEM values were <SD/2 for each self-care scale, suggesting an acceptable measurement error; in the SDC testing we provided the following reference points for a meaningful change in the self-care scales: 6.66 for self-care maintenance, 7.78 for self-care monitoring, and 7.89 for the self-care management scale. Therefore, we can assume that the inventory is accurate to measure self-care behaviors in chronic conditions.

### 4.5. Strengths and Limits

Our study presents several strengths. This is the first study to test SC-CII in a middle-income country, bringing significant knowledge to the literature on self-care. Additionally, we enrolled an adequate sample size of participants, and we used robust psychometric testing and a rigorous methodology to establish validity and reliability.

Despite its strengths, our study also has several limitations. First, we enrolled a convenience sample. We tried to overcome this limit by enrolling people of different sexes and ages, with a number of various chronic diseases, and from several centers in central and south Albania. However, the SC-CII needs further testing in other middle- and low-income countries to confirm or reject the findings of this study.

### 4.6. Implication for Clinical Practice and Research

The Albanian SC-CII is a reliable and valid instrument to use in clinical practice. Clinicians can use the SC-CII to identify the self-care behaviors of people with chronic illness and use that information to improve patients’ and their caregivers’ knowledge regarding healthy lifestyles and symptoms of diseases, and teach them appropriate management skills, or to monitor variations in the patient’s ability to self-care over time. Additionally, healthcare professionals can use this scale to compare general self-care behaviors of patients with behaviors regarding specific chronic diseases, and, therefore, they will be able to integrate the plan of care for the patient better and define appropriate interventions.

## 5. Conclusions

The Self-Care in Chronic Illness Inventory (SC-CII) is a theory-based, valid, and reliable instrument to measure self-care behaviors in the chronic illness adult population. The psychometric characteristics tested in this study supported the validity and internal consistency of the SC-CII scales. The Albanian SC-CII fills an important gap in the literature and clinical practice. Clinicians now have an instrument available to understand and improve the self-care of patients with chronic conditions in Albania.

## Figures and Tables

**Figure 1 ijerph-20-04714-f001:**
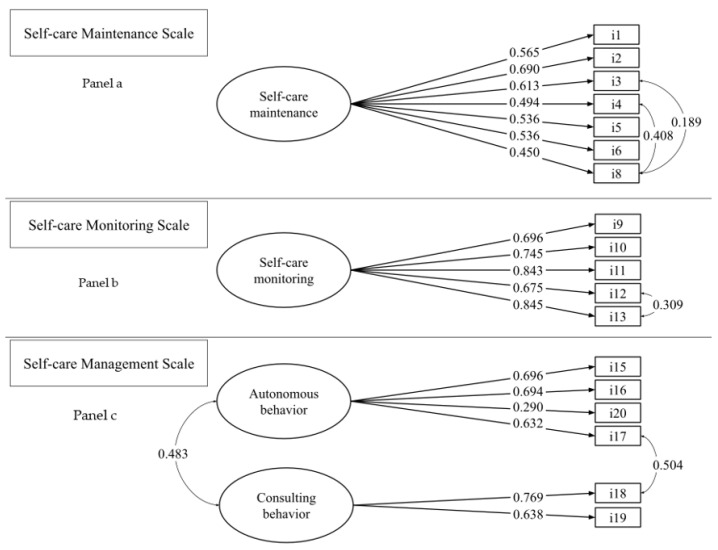
Results of the confirmatory factor analysis (*N* = 250). Note. Confirmatory factor analysis tested separately for the three scales. The first image illustrates CFA testing of the self-care maintenance scale with all seven items loaded on a single dimension. The second image illustrates CFA of testing the self-care monitoring scale with all five items loaded on a single dimension. The third image illustrates CFA testing of the self-care management scale with the six items loaded on two dimensions. Items are numbered in order of appearance in the scale. Factor loadings are represented with standardized coefficients, and are all statistically significant (*p* < 0.05 or below). Note that the two dimensions of the self-care management scale (autonomous behaviour and consulting behaviour) are not intended as subscales.

**Table 1 ijerph-20-04714-t001:** Socio-demographic and clinical characteristics of the sample (*n* = 250).

Variable	*N* (%)	Mean ± SD
**Age**		73.4 (6.4)
**Sex**		
Male	134 (53.6)	
Female	116 (46.4)	
**Marital status**		
Married/partnered	196 (78.4)	
Single	3 (1.2)	
Widow/divorced	51 (20.4)	
**Level of education**		
0–8 years	155 (62)	
≥9 years	95 (38)	
**Employment status**		
Employed	9 (3.6)	
Unemployed/retired	241 (96.4)	
**Perceived income adequacy**		
Less than needed	16 (6.4)	
Enough for living	200 (80.0)	
More than needed	34 (13.6)	
**Number of chronic conditions**		2.5 (0.69)
**Patient chronic conditions**		
Hypertension	219 (87.6)	
Diabetes mellitus	185 (74.0)	
Heart failure	112 (44.8)	
COPD	32 (12.8)	
Kidney disease	20 (8.0)	
Arthritis	20 (8.0)	
Other	28 (11.2)	
**Living with**		
Spouse/partner	195 (78.0)	
Child	125 (50.0)	
Grandchildren	75 (30.0)	
Son/daughter-in-law	82 (32.8)	
Other	5 (2.0)	

Legend. SD: standard deviation, COPD: chronic obstructive pulmonary disease.

**Table 2 ijerph-20-04714-t002:** Descriptive analysis of the SC-CII items.

Items	M	SD	Skewness	Kurtosis
**Self-care Maintenance scale** (*N* = 250)				
1. Make sure to get enough sleep?	4.03	0.944	−0.746	0.179
2. Try to avoid getting sick (e.g., flu shot, wash your hands)?	4.41	0.729	−0.873	−0.373
3. Do physical activity (e.g., take a brisk walk, use the stairs)?	3.37	1.281	−0.206	−0.988
4. Eat a special diet?	3.45	1.177	−0.335	−0.574
5. See your healthcare provider for routine health care?	4.20	0.875	−0.608	−0.886
6. Take prescribed medicines without missing a dose?	4.74	0.476	−1.524	1.308
8. Manage stress?	2.94	1.251	0.121	−0.843
**Self-care Monitoring scale** (*N* = 250)				
9. Monitor your condition?	4.11	0.881	−0.707	−0.133
10. Pay attention to changes in how you feel?	3.98	0.899	−0.420	−0.610
11. Monitor for medication side effects?	4.00	0.944	−0.452	−0.907
12. Monitor whether you tire more than usual doing normal activities?	3.96	0.956	−0.619	−0.196
13. Monitor for symptoms?	3.96	0.971	−0.528	−0.662
14. If you had symptoms in the past month, how quickly did you recognize it as a symptom of your illness?	2.85	1.473	−0.001	−1.058
**Self-care Management scale** (*N* = 226)				
15. When you have symptoms, how likely are you to … change what you eat or drink to make the symptom decrease or go away?	3.62	1.006	−0.200	−0.487
16. Change your activity level (e.g., slow down, rest)?	3.90	0.972	−0.556	−0.162
17. Take a medicine to make the symptom decrease or go away?	4.28	0.869	−0.871	−0.372
18. Tell your healthcare provider about the symptom at the next office visit?	4.39	0.783	−1.044	0.156
19. Call your healthcare provider for guidance?	3.92	1.136	−0.832	−0.144
20. Think of a treatment you used the last time you had symptoms. Did the treatment you used make you feel better?	3.38	1.078	−0.236	−0.354

Legend. SD: standard deviation.

**Table 3 ijerph-20-04714-t003:** Bivariate correlation between Self-Care of Chronic Illness Inventory, depression, and health-related quality of life (*N* = 250).

	1	2	3	4	5	6	7
1. Self-care maintenance	-	0.500	0.561	0.600	−0.289	0.487	0.325
2. Self-care monitoring		-	0.631	0.668	−0.323	0.567	0.442
3. Self-care management			-	0.696	−0.245	0.557	0.435
4. Self-care self-efficacy				-	−0.245	0.664	0.567
5. PHQ-9					-	−0.657	−0.456
6. MCS						-	0.776
7. PCS							-

MCS: mental health-related quality of life; PHQ-9: 9-Item Patient Health Questionnaire; PCS: physical health-related quality of life. Note. All correlations are statistically significant (*p* < 0.001).

**Table 4 ijerph-20-04714-t004:** Comparisons between male and female patients, dyadic care types of the Dyadic Symptom Management Scale, and Self-Care of Chronic Illness Inventory (known groups validity).

	Total Sample	Female Patient (*N* = 116, 46.6%)	Male Patient (*N* = 134, 56.6%)	T Test *p*-Value	Patient-Oriented Dyadic Care Type (*N* = 38, 15.2%)	Caregiver-Oriented Dyadic Care Type (*N* = 31, 12.4%)	*T* Test *p*-Value
	M (SD)	M (SD)	M (SD)		M (SD)	M (SD)	
Self-care maintenance (N = 250)	72.0 (16.0)	69.7 (15.4)	74.1 (16.3)	0.004	75.4 (15.5)	55.8 (11.6)	0.002
Self-care monitoring (N = 250)	75.0 (19.1)	73.5 (17.8)	76.3 (20.3)	0.002	79.2 (18.4)	61.5 (19.5)	0.003
Self-care management (N = 226)	72.4 (15.9)	72.5 (14.8)	72.1 (16.9)	0.006	68.8 (16.7)	61.6 (16.5)	0.004

M: mean; SD: standard deviation.

## Data Availability

The data from this study are available upon reasonable request addressed to the authors.

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
