# Peer review of "Psychometric Characteristics of the Self-Care of Chronic Illness Inventory in Older Adults Living in a Middle-Income Country"

_ijerph, 2023, doi:10.3390/ijerph20064714_

Round 1

Reviewer 1 Report

This is a lot of work done in a thorough and conscientious way, which gives it great value in middle-income countries in South East Europe.

It is an instrument that can have a great application for patients with chronic diseases to promote self-care in its different dimensions.

It will be an article that will have a great impact on the care of the elderly, in the geriatric consultation and in chronic patients with various conditions.

We noticed some  improvements that are signaled in the attachment: 

  • Line 77 where it states: migration to other nations or versus, i would suggest “migration to other nations or to urban areas”

  • Line 82, who has employed the different approaches, I would add it. (ex. In the last decade, the govt has employed…)

  • Line 86, “additionally” would better be replaced for “consequently,”

  • Line 87, if “however” was added before “despite this efforts” so it would read: “However, despite these efforts” the sentences could be read more cohesively

  •  Line 92, a comma is missing after: in Albania, long term…

  • Lines 98 and 99 the word “in” could be deleted for a better translation: “many languages such as Arabic… and then: but not Albanian”

  • Line 104, the apostrophes do not match in the parentheses

  • Line 127, there is a typo in the word old, it reads “olde”

Educating in health and self-care is one of the great challenges of current medicine for middle-income health systems that want to optimize their resources.

We hope that the authors continue working in this line of research that will bear great fruit in the coming years.

In relation to the methodology, a very good job has been done, worthy of recognition.

Very opportune to point out what is reported by the literature on the subject at present and explain the differences in relation to cultural and social aspects.

We look forward to seeing your post soon.

Author Response

Thank you so much for taking your time on our manuscript and for helping us make the manuscript clearer. Thank you for identifying typos and unclear periods. We admitted all your suggestions and performed changes to the manuscript.

Reviewer 2 Report

Dear authors

Congrats for your work. i have  more to suggest than try to be more concise in your introduction and in the description of the instruments. 

The manuscript is not allways easy to read because some parts are dense, like the presentation of the results because you duplicate some information of the tables in the text, and the tables are not allways easy to read.

In general the article is good, and most of all, has an implication for clinical practice and this is waht really matters. 

Great job

Author Response

Thanks so much for your review of our manuscript. As suggested, we summarized the description of the instruments section and improved the table 1.
